# Intercostal Muscles Oxygenation and Breathing Pattern during Exercise in Competitive Marathon Runners

**DOI:** 10.3390/ijerph18168287

**Published:** 2021-08-05

**Authors:** Felipe Contreras-Briceño, Maximiliano Espinosa-Ramírez, Eduardo Moya-Gallardo, Rodrigo Fuentes-Kloss, Luigi Gabrielli, Oscar F. Araneda, Ginés Viscor

**Affiliations:** 1Laboratory of Exercise Physiology, Department of Health Science, Faculty of Medicine, Pontificia Universidad Católica de Chile, Av. Vicuña Mackenna 4860, Santiago 7820436, Chile; maespinosa@uc.cl (M.E.-R.); eemoya@uc.cl (E.M.-G.); rfuentesk@uc.cl (R.F.-K.); lgabriel@uc.cl (L.G.); 2Physiology Section, Department of Cell Biology, Physiology and Immunology, Faculty of Biology, Universitat de Barcelona, Av. Diagonal 643, 08028 Barcelona, Spain; gviscor@ub.edu; 3Advanced Center for Chronic Diseases (ACCDiS), Division of Cardiovascular Diseases, Faculty of Medicine, Pontificia Universidad Católica de Chile, Av. Sergio Livingstone 1007, Santiago 8380000, Chile; 4Laboratory of Integrative Physiology of Biomechanics and Physiology of Effort (LIBFE), Kinesiology School, Faculty of Medicine, Universidad de los Andes, Av. Monseñor Alvaro del Portillo 12455, Santiago 7620001, Chile; ofaraneda@miuandes.cl

**Keywords:** exercise, runners, near-infrared spectroscopy, respiratory muscles, respiration

## Abstract

The study aimed to evaluate the association between the changes in ventilatory variables (tidal volume (Vt), respiratory rate (RR) and lung ventilation ($\dot{V}$E)) and deoxygenation of *m.intescostales* (∆SmO_2_-*m.intercostales*) during a maximal incremental exercise in 19 male high-level competitive marathon runners. The ventilatory variables and oxygen consumption ($\dot{V}$O_2_) were recorded breath-by-breath by exhaled gas analysis. A near-infrared spectroscopy device (MOXY^®^) located in the right-hemithorax allowed the recording of SmO_2_-*m.intercostales*. To explore changes in oxygen levels in muscles with high demand during exercise, a second MOXY^®^ records SmO_2_-*m.vastus laterallis*. The triphasic model of exercise intensity was used for evaluating changes in SmO_2_ in both muscle groups. We found that ∆SmO_2_-*m.intercostales* correlated with $\dot{V}$O_2-peak_ (r = 0.65; *p =* 0.002) and the increase of $\dot{V}$E (r = 0.78; *p* = 0.001), RR (r = 0.54; *p =* 0.001), but not Vt (*p =* 0.210). The interaction of factors (muscles × exercise-phases) in SmO_2_ expressed as an arbitrary unit (a.u) was significant (*p* = 0.005). At VT1 there was no difference (*p* = 0.177), but SmO_2_-*m.intercostales* was higher at VT2 (*p* < 0.001) and $\dot{V}$O_2-peak_ (*p* < 0.001). In high-level competitive marathon runners, the *m.intercostales* deoxygenation during incremental exercise is directly associated with the aerobic capacity and increased lung ventilation and respiratory rate, but not tidal volume. Moreover, it shows less deoxygenation than *m.vastus laterallis* at intensities above the aerobic ventilatory threshold.

## 1. Introduction

During the race, high-level competitive marathon runners use about 75% of the maximum oxygen consumption ($\dot{V}$O_2-máx._) and keep lung ventilation ($\dot{V}$E) near 80 L·min^−1^ [1]. This requires the respiratory muscles recruitment (e.g., *m.diaphragma, m.intercostales, m.serratus (posterior* and *anterior*)), leading to an increase in $\dot{V}$O_2_ distributed to these muscles ($\dot{V}$O_2-RM_), which in trained men has been proven to reach up to 10% [2]. This elevated energy cost of breathing (COB) requires higher local nutrients and oxygen supply, an aspect that has been reported could diminish blood flow to locomotor muscles, causing an intolerance to effort and early end of exercise (metabolic reflex) [2,3,4].

Different factors are implicated in a high COB by exercise; one of them is the breathing pattern adopted during physical exertion, understood as the relative contribution of Vt and RR to the increase in $\dot{V}$E. Few studies have evaluated how the relative changes in these ventilatory variables during exercise could affect the oxygenation of respiratory muscles, a relevant aspect because of the wide variability observed according to the type (running or cycling), duration (short or long-exercise) and intensity (steady-state or incremental) of the physical exercise [5,6,7].

Evaluating the respiratory muscle’s oxygenation level is difficult due to the complex anatomical arrangement and the considerable variation in muscular recruitment with varying degrees of ventilation. Nevertheless, a new non-invasive method has been proposed to determine the ratio between the supply-to-consumption of oxygen as reflected by muscle saturation (SmO_2_) evaluated in the *m.intercostales*, which has great importance in high intensities of exercise; this can be determined by continuous near-infrared spectroscopy (NIRS, 630–850 nm) that measures the changes in oxygenated hemoglobin (O_2_Hb) and myoglobin (mHb) at a microvascular level [8,9,10]. In sports science, the changes of SmO_2_ (deoxygenation, ∆SmO_2_) during physical effort have been extensively studied in locomotor muscles, with far less attention to respiratory muscles [11,12,13]. To our knowledge, in high-level competitive marathon runners, it is unexplored how the breathing pattern adopted during physical exertion is associated with SmO_2_-*m.intercostales*, a novel aspect requiring attention due to the possible respiratory limitations to exercise that could be modulated by respiratory muscle training.

Thus, the main goal of this study was to evaluate the association between breathing pattern and deoxygenation of the *m.intercostales* (∆SmO_2_-*m.intercostales*) during an incremental physical effort test in high-level competitive marathon runners. Secondarily, we aimed to determine the association between oxygen consumption and the relative increase of each ventilatory variable (lung ventilation ($\dot{V}$E), respiratory rate (RR) and tidal volume (Vt)), ∆SmO_2_-*m.intercostales* and ∆SmO_2_*-m.vastus laterallis*. Furthermore, we explore the possible limitation to exercise by comparing the changes between ∆SmO_2_-*m.intercostales* and ∆SmO_2-_*m.vastus laterallis* during exercise protocol according to the triphasic exercise intensity model determined by the ventilatory thresholds (ventilatory threshold 1 (VT1) and ventilatory threshold 2 (VT2)).

## 2. Materials and Methods

### 2.1. Design of Study and Participants

A cross-sectional observational study that assessed 19 healthy male high-level competitive marathon runners (age 22.9 ± 1.9 years; height 173.1 ± 4.2 cm; weight 66.5 ± 6.7 kg; body mass index (BMI) 22.2 ± 2.8 kg·m^−2^; peak oxygen consumption ($\dot{V}$O_2-peak_) 62.3 ± 4.6 mL·min^−1^·kg^−1^; training volume 125.1 ± 31.4 km·week^−1^, best personal time obtained in a marathon (during the last 3 years) 157 ± 23 min) without a history of systemic problems, such as respiratory, cardiovascular, metabolic, musculoskeletal or neoplastic diseases, or any infectious or inflammatory process, for at least two weeks before measurements (recruitment by convenience). The participants did not consume drugs, antioxidants or any nutritional support. The sample size calculation was done by the software G*Power^®^ 3.1 (Heinrich-Heine-University, Dusseldorf, Germany) using previous data concerning the association found between SmO_2_-*m.intercostales* and $\dot{V}$O_2_ in runners with similar characteristics to participants in this study (r = 0.64; *p* = 0.001), considering a significance level of 5%, power of 80% and a two-tail test, plus 10% of data losing. All the participants were informed of the purpose, protocol and procedures before informed consent was obtained from them. This study was approved by the ethics committee of the Pontificia Universidad Católica de Chile (Institutional Review Board, protocol number 180305007, date of approval: 5 June 2018). The study was carried out according to the Declaration of Helsinki for human experimentation.

### 2.2. Procedures

The evaluations were done in the Exercise Physiology Laboratory of the Pontificia Universidad Católica de Chile under controlled environmental conditions (room temperature, 20 ± 2 °C; relative humidity, 40 ± 2%) and fixed schedule (9:00 to 14:00 h). All the participants were instructed to avoid physical activity 24 h before the day of measurement and not to engage in alcohol, caffeine or other stimulants and food intake for at least three hours before the tests.

### 2.3. Ergospirometry

The aerobic capacity was assessed by determining the $\dot{V}$O_2-peak_. This test was performed on a treadmill ergometer (HP Cosmos, Traunstein, Germany) in an incremental exercise until voluntary exhaustion, despite verbal stimuli (respiratory quotient, 1.20 ± 0.05). The exercise protocol consisted of a 3-min rest, 5-min warm-up (8 km·h^−1^) and subsequent increase of 2 km·h^−1^ every 150 s, until all criteria for stopping the test were met. The treadmill slope throughout the test was fixed at 2%. Heart rate, fingertip oxygen saturation and ventilatory variables ($\dot{V}$E, RR, Vt) were continuously recorded. Ventilatory and spired gas composition data ($\dot{V}$O_2_, $\dot{V}$CO_2_, $\dot{V}$E, RR and Vt) were *breath-by-breath* obtained using an ergospirometer (MasterScreen CPX, Jaeger™, Traunstein, Germany) and expressed under standard temperature pressure dry air (STPD).

### 2.4. Triphasic Model of Exercise Intensity Determined by Ventilatory Thresholds

To analyze the changes in the variables measured (SmO_2_, $\dot{V}$E, RR and Vt) according to the triphasic model, the ventilatory 1 (VT1) and 2 (VT2) and $\dot{V}$O_2-peak._ were determined by two experienced and blinded evaluators using the visual method [14]. The VT1 was the initial departure from $\dot{V}$E linearity, the beginning of a systematic increase in the ventilatory equivalent of O_2_ ($\dot{V}$E·$\dot{V}$CO_2_^−1^) and the end-tidal pressure value of O_2_ (Pet-O_2_). The VT2 was the secondary increase in $\dot{V}$E, $\dot{V}$E·$\dot{V}$O_2_^−1^, a marked increase in the ventilatory equivalent of CO_2_ ($\dot{V}$E·$\dot{V}$CO_2_^−1^) and a decrease in the pressure value at the end of CO_2_ expiration (Pet-CO_2_) during exercise, above VT1 [15]. In case of a discrepancy between the two evaluators, the opinion of an experienced third blinded evaluator was possible, accepting as the definitive criterion that point at which at least two evaluators agreed [16]. However, none of the cases revealed differences between the two evaluators in this study. Regarding $\dot{V}$O_2-peak_, the highest value of the last 30 s obtained during the incremental maximum effort test was considered, based on established criteria for the determination [17].

### 2.5. Measurement of SmO_2_

During the protocol, muscle oxygen saturation (SmO_2_) was non-invasively evaluated using the monitor MOXY^®^ (Fortiori Design LLC, Hutchinson, MN, USA), which emits light waves close to the infrared range (near-infrared spectroscopy, NIRS (630–850 nm)) from diodes to the surrounding tissue and records total hemoglobin (THb) and myoglobin (mHb) concentrations at the microvascular level [9]. This device records the amount of light that returns to two detectors placed 12.5 and 25.0 mm from the source, thus locally recording SmO_2_ through the interpretation of THb and mHb levels [18]. The SmO_2_ data were recorded using the PeriPedal^®^ software (PeriPedal, Indianapolis, IN, USA) with a sampling frequency of 2 Hz [19]. The light penetration depth is half of the distance between the emitting source and the detector (±12 mm) [19]. To determine the SmO_2_-*m.intercostales*, a MOXY^®^ device was located in the seventh intercostal space of the anterior axillary line of the right hemithorax [20]. A second MOXY^®^ device was placed 5 cm lateral to the midpoint of the imaginary line between the upper edge of the patella and the greater trochanter of the right femur [20]. The position of both MOXY^®^ devices was attached to the skin using the material suggested by the manufacturer, in addition to extra fixation with a cohesive band on the surroundings of the measurement zone, avoiding excessive compression that could alter the SmO_2_ record (similar to that used in our previous study) [20]. Figure 1 illustrates the positions of the NIRS devices.

### 2.6. Breathing Pattern

To facilitate the interpretation of the breathing pattern of each participant (corresponding to changes in ventilatory variables ($\dot{V}$E, RR and Vt) during the exercise protocol), a standardization of the obtained values was performed by transferring them to arbitrary relative units considering the number of times that each variable increased from rest (taken as the unit (1.0) reference value) to $\dot{V}$O_2-peak_ phases (n° times = (ventilatory variable value at $\dot{V}$O_2-peak_ phase–ventilatory variable value at rest phase) ventilatory variable value at rest phase^−1^).

### 2.7. Data Analysis

The registration of the variables monitored by using the ergoespirometer and the SmO_2_ devices was manually synchronized by two operators. Thus, each participant had an initial record of 90 s in the bipedal position, followed by 180 s corresponding to the resting phase. The values of ventilatory variables ($\dot{V}$E, RR and Vt) and SmO_2_ considered for calculations correspond to the average of the last 30 s of each phase recordings. The maximal deoxygenation (∆) corresponds to (SmO_2_ at rest-SmO_2_ at $\dot{V}$O_2-peak_) · (SmO_2_ at $\dot{V}$O_2-peak_)^−1^ (%). To compare the SmO_2_ in the muscle groups assessed during the exercise protocol, the data were expressed as an arbitrary unit (a.u), where the value of SmO_2_ obtained at the rest phase was considered the maximum or 1, values obtained in other phases were expressed as a ratio (e.g., SmO_2_-*m.intercostales* at VT1 (a.u) = SmO_2_ at VT1 (%) · SmO_2_ at rest^−1^ (%)). These methods of expressing muscle oxygenation are used because SmO_2_ values present variability between muscle groups mainly since their analysis includes both hemoglobin and myoglobin quantification (myoglobin level has been reported to correspond to between 30% to 50% of the signal at different muscles) [21], whose concentration is variable depending on the morphofunctional and structural characteristics (such as capillarization and skeletal muscle fiber type composition) of the different muscle groups in each athlete.

### 2.8. Statistical Analysis

Data normality was checked using the Shapiro–Wilk test. The association of $\dot{V}$O_2-peak_, the relative increase of $\dot{V}$E, RR, Vt (breathing pattern) and ∆SmO_2_ at *m.intercostales* and *m.vastus laterallis* were assessed using the Pearson correlation coefficient. A two-way ANOVA test allowed the comparison of the SmO_2_ (a.u) levels between muscles assessed among different exercise protocol phases (VT1, VT2 and $\dot{V}$O_2-peak_) when the interaction of the factors (muscles x exercise phases) was significant (*p* < 0.05). Subsequent multiple comparisons were analyzed using the Tukey post-hoc test. The level of significance for all analyses was *p* < 0.05. The statistical software used was GraphPad Prism 7.0 (San Diego, CA, USA).

## 3. Results

The values of SmO_2_ (%) (*m.intercostales* and *m.vastus laterallis*), total hemoglobin (THb) and ventilatory variables ($\dot{V}$E, RR and Vt) are shown in Table 1.

### 3.1. Correlations

The $\dot{V}$O_2-peak_ had a moderate association with the number of times that $\dot{V}$E (r = 0.66, *p* = 0.001) and RR (r = 0.53, *p* = 0.001), but not with changes in Vt (*p* = 0.540) (Figure 2).

The ∆SmO_2_-*m.intercostales* had a moderate association with $\dot{V}$O_2-peak_ (r = 0.65, *p* = 0.002) and ∆SmO_2_-*m.vastus laterallis* (r = 0.61, *p* = 0.005). The ∆SmO_2_-*m.vastus laterallis* correlated with $\dot{V}$O_2-peak_ (r = 0.61, *p* = 0.005) (Figure 3).

Finally, ∆SmO_2_-*m.intercostales* had a high association with the number of times that $\dot{V}$E (r = 0.78, *p* = 0.001) increased, and moderate with RR (r = 0.54, *p* = 0.001) but did not correlate with changes in Vt (*p* = 0.210) (Figure 4).

### 3.2. Triphasic Model of Exercise Intensity and SmO_2_

The interaction of factors (muscles X protocol phases) in SmO_2_ (a.u) was significant (*p* = 0.005), at VT1 were not different (*p* = 0.177), but SmO_2_-*m.intercostales* was higher at VT1 (*p* < 0.001) and $\dot{V}$O_2-peak_ (*p* < 0.001). Regarding comparison to rest phase, the SmO_2_-*m.intercostales* was different in the VT2 (*p* < 0.001) and $\dot{V}$O_2-peak_ (*p* < 0.001), but not in VT1 (*p* = 0.122). While in SmO_2_-*m.vastus laterallis*, rest was different in VT1 (*p* = 0.001), VT2 (*p* < 0.001) and $\dot{V}$O_2-peak_ (*p* < 0.001) (Figure 5).

## 4. Discussion

The purpose of this study was to evaluate the association between deoxygenation of *m.intercostales* (∆SmO_2_*-m.intercostales*) and breathing pattern adopted during maximal effort in competitive marathon runners. Our main finding is that ∆SmO_2_-*m.intercostales* was associated with the number of times that the $\dot{V}$E and RR increased (Figure 4a,b), but not with Vt changes (Figure 4c). This implies that a breathing pattern determined mainly by RR increase induces higher deoxygenation in *m.intercostales*. We observed that relative changes in $\dot{V}$E, RR and ∆SmO_2_*-m.intercostales* were directly associated with $\dot{V}$O_2-peak_ (Figure 2a,b and Figure 3a). In other words, as expected, as lung ventilation increases, the cost of breathing and oxygen consumption also rises, thus favoring the process of lung diffusion and blood transport of oxygen to the tissues [1].

Concerning the associations found, it has been reported that a breathing pattern driven by higher RR leads to a more significant work of breathing (WOB) than increasing Vt [22]. This seems likely because it has been shown that RR linearly rises with the increase of intensity of exercise, which gives it a fundamental role in the evaluation and prescription of physical exercise, given its high correlation with the *Perceived Effort Scale* (RPE, r = 0.74) [23]. Physiologically, this aspect could be explained because at a high intensity of exercise, once Vt reaches 50–60% of the vital capacity, the increase in $\dot{V}$E could be necessarily achieved mainly by increasing RR, thus implying an exponential increase in work of breathing above VT2, due to the lung mechanical compliance that limits the increase in Vt. It is also well known that after VT2, recruitment of the respiratory accessory muscles starts. Therefore, it seems plausible that SmO_2_ will decrease, thus showing a tight association with the increase in RR [3]. Thereby, at high exercise intensities, after the exponential increase in RR, there is an increased imbalance in the oxygen supply/consumption relationship into accessory respiratory muscles, demonstrated by the direct association between ∆SmO_2_-*m.intercostales* and the number of times that RR increased.

We also found that the maximum changes (∆SmO_2_) in the *m.intercostales* and *m.vastus laterallis* during exercise directly correlate with $\dot{V}$O_2-peak_ (Figure 3a,b). Therefore, athletes who showed better aerobic capacity were those who extracted more O_2_ at the muscle levels, reaching the highest energy costs in both locomotor and respiratory muscles (∆SmO_2_*-m.vastus laterallis* and ∆SmO_2_*-m.intercostales*, respectively). These findings agree with those previously reported separately in locomotor and respiratory muscles [18]. Moreover, we found a moderate association of maximum SmO_2_ changes between muscle groups (Figure 3c). This is consistent with our previous results, where we found a direct association between SmO_2-_*m.intercostales* and SmO_2-_*m.vastus laterallis* in marathon runners, although in such a study, the variables were presented as absolute values (%) and not as relative maximal changes (∆SmO_2_) [20]. Furthermore, this seems to be consistent with the physiological responses reported in competitive athletes. In this regard, it has been discussed that in athletes, in addition to central cardiovascular adaptations, other peripheral adaptations are present, such as increased conduit artery size, including enlargement of epicardial arteries and those supplying skeletal muscles [24]. However, this appears to be a systemic adaptation more than a local effect, and further studies are required that could help to explain why, in marathon runners, the ∆SmO_2_*-m.intercostales* and ∆SmO_2_*-m.vastus laterallis* are so closely associated at high-intensity exercise levels.

Interestingly, the oxygenation decrease was more significant in locomotor muscles than in respiratory muscles (Figure 5). Therefore, deoxygenation of respiratory muscles does not appear to be a limiting factor of maximal exercise, although it could compete for oxygen supply and thus restrict to some extent a better O_2_ supply/consumption relationship of great locomotory groups, such as *m.vastus laterallis*, during physical exercise. This approach, and its quantification, requires further study since it has been shown that elite marathon runners present a respiratory limitation to exercise (expiratory flow limitation and high WOB levels) [1,25], which could be reflected in higher ∆SmO_2_*-m.intercostales*. Unfortunately, this study’s lack of local blood flow measurements does not allow us to identify whether ∆SmO_2_ variation could be due to the local O_2_ delivery or consumption components.

Another novel result in this study is that SmO_2_ and THb in both muscle groups showed similar results. SmO_2_ decreased in VT1, VT2 and $\dot{V}$O_2-peak_ phases compared to rest levels, whereas THb did not change (Table 1). These data are consistent with previous reports, confirming the replicability of our exercise protocol [20]. It could be hypothesized that the change in breathing pattern during an incremental exercise in our participants, both before and after reaching VT2, could optimize the ∆SmO_2_*-m.intercostales*. Recent reports argue that among the multiple stimuli that cause the increase in $\dot{V}$E is the involvement of a “central command” that would allow voluntary control of RR and Vt ventilatory pattern changes. This ability can be trained, ensuring a better O_2_ delivery/demand related to the locomotor musculature to improve aerobic capacity and sports performance by adjusting the changes in ventilatory response according to the exercise modality [26,27].

On the other hand, respiratory muscle training could also be an option to optimize oxygenation of respiratory muscles by decreasing ∆SmO_2_*-m.intercostales*. Concerning this, it has been reported in patients with chronic heart failure (HF) that eight weeks of respiratory muscle training reduced deoxygenation at *m.intercostales* [28]. A possible reason for these results could be attributed to the pathophysiological context of the high basal COB of these patients. In athletes, similar results were found in cyclists after an inspiratory muscle training (IMT), but not during endurance exercise, only at moderate-to-heavy inspiratory loading trials [29]. This study in professional women football players found a decrease in *m.intercostales* oxygenation levels during an incremental exercise after six weeks of IMT [30]. According to this evidence, it could be interesting to investigate the effect of respiratory training on the ∆SmO_2_*-m.intercostales* during exercise, both in clinical and sports environments employing devices based on different rationale (e.g., inspiratory thresholds, Spirotiger^®^, Chamonix-Mont-Blanc, French), to explore changes in the breathing patterns adopted during a physical effort that could influence on the physical performance.

As a limitation of this study, we declare the non-evaluation of adipose tissue at the sites where the portable NIRS devices were placed. Therefore, it was impossible to determine the precise depth of the muscle groups evaluated, making it challenging to assess SmO_2_. However, our participants were athletes with a BMI of 22.2 ± 2.8 kg·m^−2^ and we consider they have a low percentage of subcutaneous adipose tissue, thus minimizing the possibility of incorrect SmO_2_ lectures. Furthermore, we consider that it would have been desirable to record oxygenated and deoxygenated hemoglobin levels, thus allowing us to assess the blood flow in the involved muscles. This could help us know if the changes at SmO_2_ were a consequence of excessive oxygen demand or insufficient delivery or both.

In future studies, we believe it is essential to incorporate sportswomen as subjects. It is unknown whether there are differences in ∆SmO_2_*-m.intercostales* in comparison to men. Although there are data that report sex differences when assessing $\dot{V}$O_2_ at the level of respiratory muscles [30], women showed higher values at submaximal and maximal intensity. Unfortunately, the protocol applied to evaluate respiratory oxygen consumption was not the same as the one used in the present study, preventing extrapolating the results.

## 5. Conclusions

The deoxygenation of *m.intercostales* during incremental maximal exercise in high-level competitive marathon runners is directly associated with the respiratory rate but not with changes in tidal volume, thus suggesting that the breathing pattern adopted during physical exercise could play an important role in athletic performance. The deoxygenation of the *m.vastus laterallis* was more significant than *m.intercostales* at intensities above the aerobic ventilatory threshold. This aspect should be investigated by the possible implicated effect of metabolic reflex as a limitation to exercise progression. This study also ratifies the use of portable NIRS devices in the accessory respiratory muscles as a novel way to quantify the local oxygen supply/consumption relationship during maximal effort.

## Figures and Tables

**Figure 1 ijerph-18-08287-f001:**
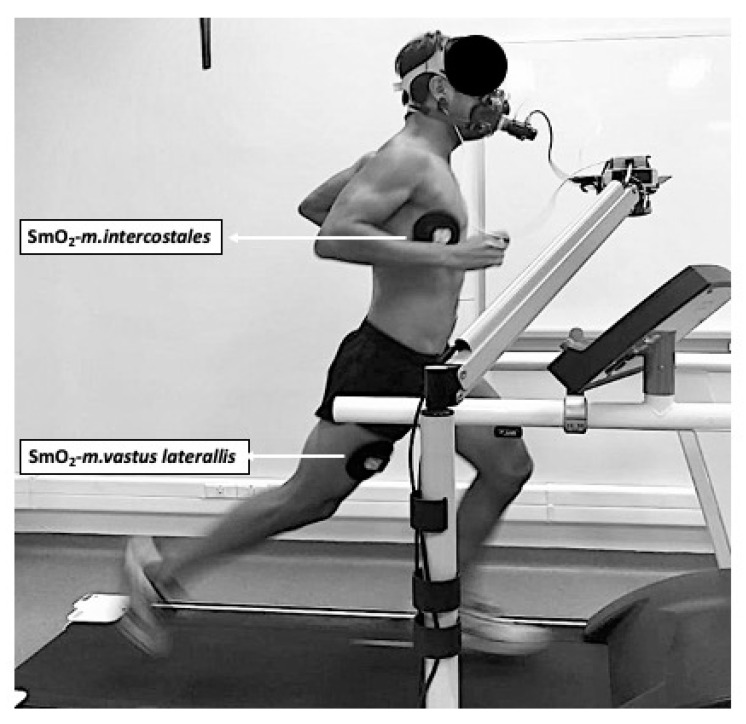
Placement of the muscle oximetry NIRS devices in the thorax (*m.intercostales*) and leg (*m.vastus laterallis*).

**Figure 2 ijerph-18-08287-f002:**
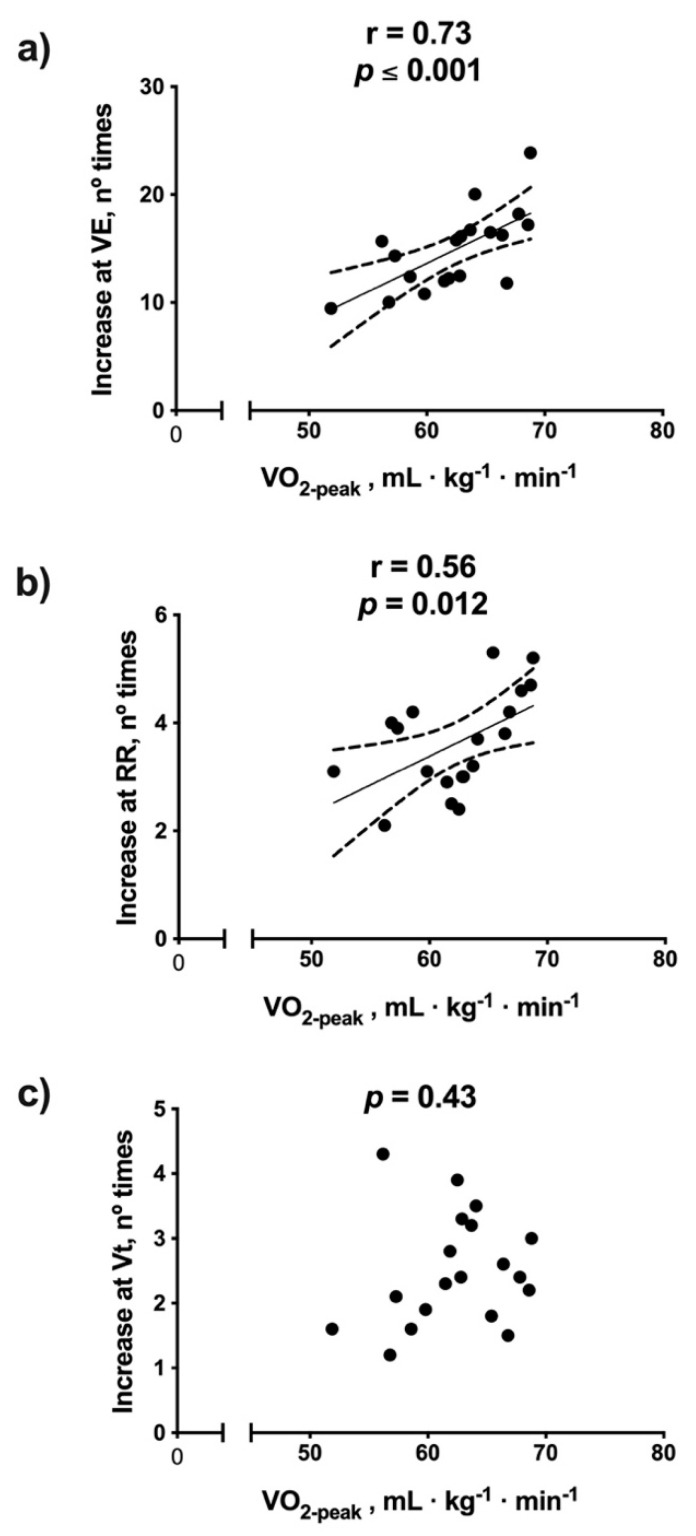
Association between $\dot{V}$O_2_ (oxygen consumption) and the number of times that: (**a**) $\dot{V}$E (lung ventilation), (**b**) RR (respiratory rate) and (**c**) Vt (tidal volume).

**Figure 3 ijerph-18-08287-f003:**
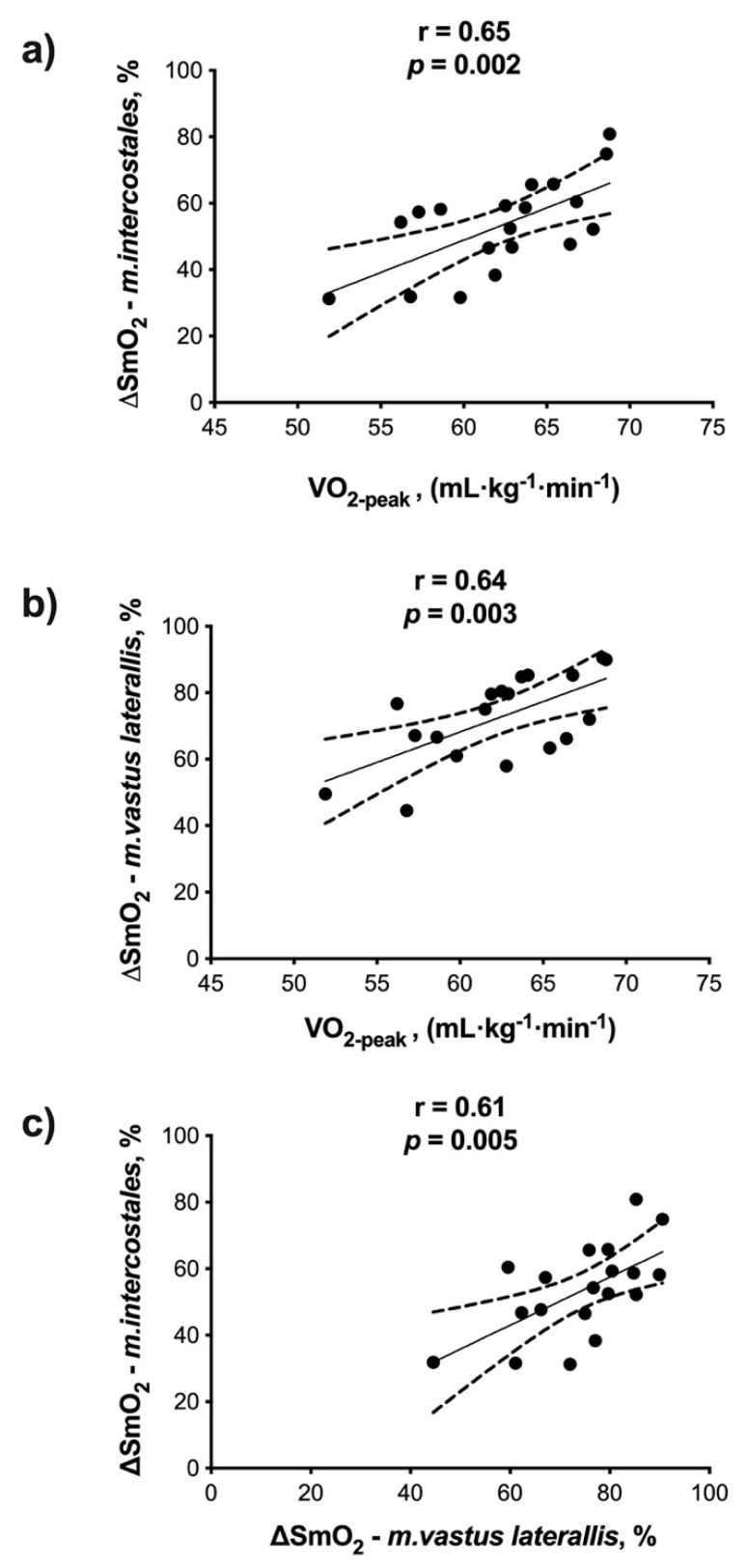
Association between $\dot{V}$O_2_ (oxygen consumption) and: (**a**) ∆SmO_2_-*m.intercostales* and (**b**) ∆SmO_2_*-m.vastus laterallis*. (**c**) Association between ∆SmO_2_-*m.intercostales* and ∆SmO_2_*-m.vastus laterallis*.

**Figure 4 ijerph-18-08287-f004:**
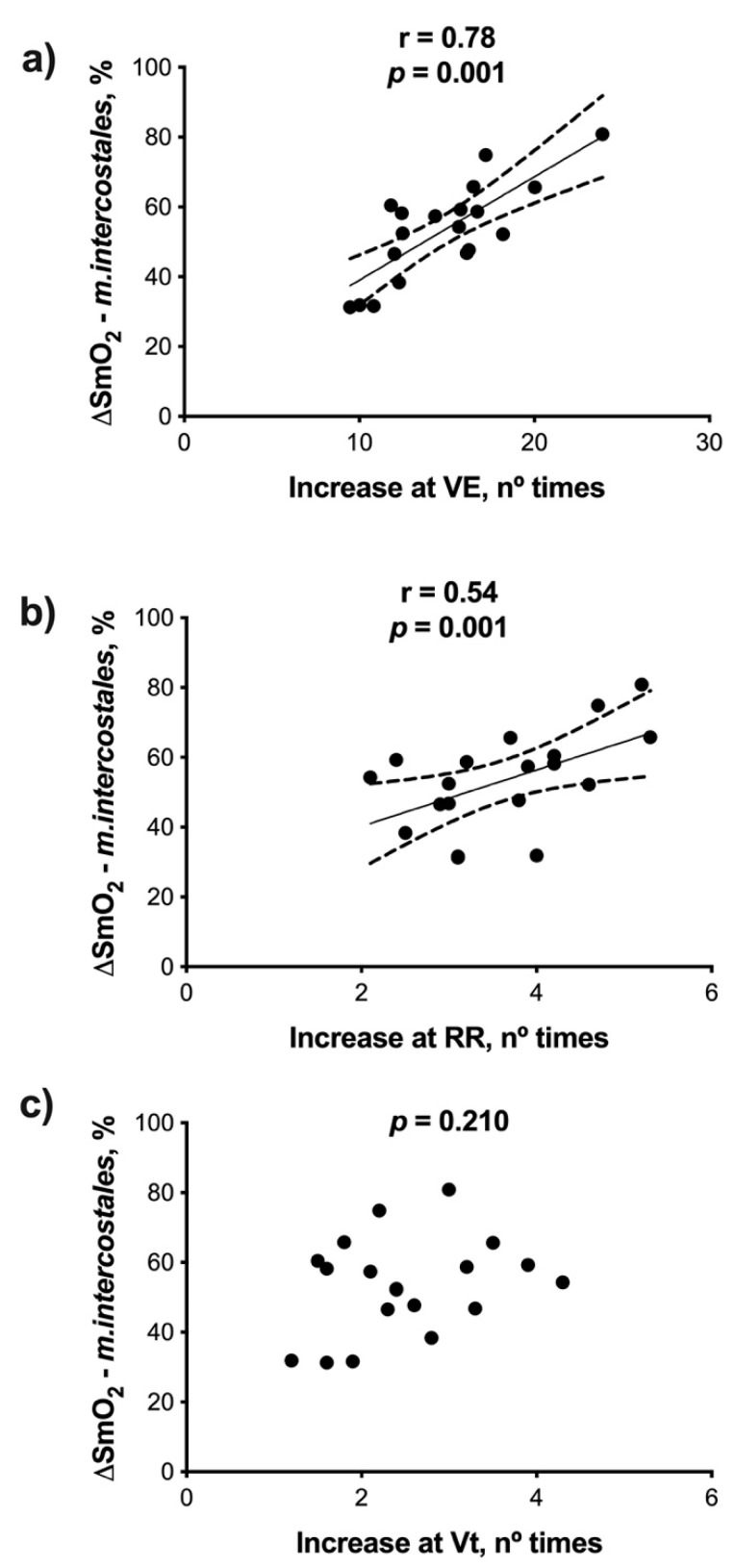
Association of ∆SmO_2_-*m.intercostales* with the number of times that: (**a**) $\dot{V}$E (lung ventilation), (**b**) RR (respiratory rate) and (**c**) Vt (tidal volume) increased.

**Figure 5 ijerph-18-08287-f005:**
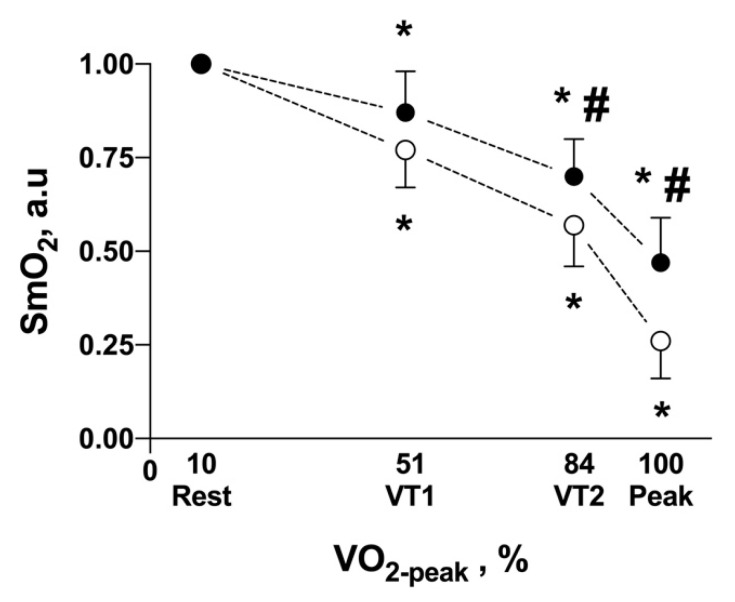
Oxygen saturation (SmO_2_) in respiratory (*m.intercostales*, solid circles) and locomotor muscles (*m.vastus laterallis,* hollow circles) in the triphasic model of exercise intensity. Values are expressed as an arbitrary unit (a.u). * *p* < 0.05 respect to rest-phase; # *p* < 0.05 comparison between muscles.

**Table 1 ijerph-18-08287-t001:** Oxygen consumption, muscle oxygen saturation, total hemoglobin and ventilatory variables in the different phases of exercise (n = 19).

	Phases			
Variables	Rest	VT1	VT2	$\dot{V}$O_2-peak_
$\dot{V}$O_2_ (mL·kg^−1^·min^−1^)	6.35 ± 0.75	31.30 ± 4.15 *	52.01 ± 3.91 *	62.32 ± 4.65 *
SmO_2−_*m.intercostales* (%)	74.6 ± 10.7	65.1 ± 12.6 *	52.9 ± 11.4 *	35.2 ± 12.8 *
∆SmO_2−_*m.intercostales* (%)	-	-	-	53.4 ± 13.9
SmO_2−_*m.vastus lateralis* (%)	63.2 ± 10.9	48.7 ± 9.6 *	36.1 ± 10.2 *	17.1 ± 7.5 *
∆SmO_2−_*m.vastus lateralis* (%)	-	-	-	72.4 ± 13.5
THb *m.intercostales* (g·dL^−1^)	12.4 ± 0.5	12.4 ± 0.5	12.4 ± 0.5	12.3 ± 0.6
THb *m.vastus laterallis* (g·dL^−1^)	12.6 ± 0.4	12.5 ± 0.5	12.4 ± 0.5	12.6 ± 0.6
$\dot{V}$E (L·min^−1^)	10.4 ± 2.5	43.2 ± 10.2 *	99.5 ± 18.1 *	158.8 ± 18.9 *
RR (bpm)	13.1 ± 1.7	25.3 ± 6.7 *	40.7 ± 7.6 *	59.3 ± 8.4 *
Vt (L)	0.81 ± 0.20	1.75 ± 0.41 *	2.47 ± 0.43 *	2.71 ± 0.30 *

Data are presented as mean ± standard deviation. *: *p* < 0.05 (R.M-ANOVA-test, statistically difference with the previous phase) $\dot{V}$O_2_: oxygen consumption; SmO_2_: muscle oxygen saturation; ∆SmO_2_ (%): ((SmO_2_ at rest phase-SmO_2_ at $\dot{V}$O_2-peak_ phase) (SmO_2_ in $\dot{V}$O_2-peak_ phase)^−1^) × 100; THb: total hemoglobin; $\dot{V}$E: lung ventilation; RR: respiratory rate; Vt: tidal volume.

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
