# Peer review of "Intercostal Muscles Oxygenation and Breathing Pattern during Exercise in Competitive Marathon Runners"

_ijerph, 2021, doi:10.3390/ijerph18168287_

Round 1
Reviewer 1 Report
Line 38: Based on the content, the authors might mean 75% of "VO2max (or VO2peak)".
Figure 5: There are two groups of muscles represented by solid and empty dots in the figure. However, the author did not define what sloid and empty dots represent. I strongly recommend including this information in the figure description.
Line 277: May need to improve the grammar of this sentence.
Reviewer 2 Report
After reading the work, there are some concerns that should be addressed in order to increase the quality of the paper.
21. What do you mean by competitive? Authors should specify if they are elite, high-level or amateur competitors.
30. Did you mean "At VT1 there were no..."?
38. This sentence is incorrect. Did authors mean VO2max? That would make more sense. Furthermore, the sentence shuold be more accurate. These values would be for any kind of marathon runners? Or would it depend on the training level?
46. Did you mean "one of them"
82. Indicate the competition level of the subjects.
108. Why the slope was set at 2% instead of 1%? Please indicate
The presentation of table 1 is very bad. Authors must present that table more clearly with each value using one line rather than 2.
Why authors don't show the tendency line in figure 2 c)
Reviewer 3 Report
GENERAL COMMENTS
The study titled “Intercostal Muscles Oxygenation and Breathing Pattern during Exercise in Competitive Marathon Runners” investigated the association between the changes in ventilatory variables 19 (tidal volume, respiratory rate, and lung ventilation) and deoxygenation of intercostal muscles during a maximal incremental exercise in nineteen male competitive 21 marathon runners. This paper deals with an issue not frequently addressed in the scientific literature and the results shown are of particular interest for the sports scientists. Moreover, the study has overall solid design and is concisely written. However, there are still few things that need to be corrected.
SPECIFIC COMMENTS
Abstract
Page 1; line 20-21: Change “m.intescostales” for “intercostal muscles”.
I think that to be more accurate use this abbreviation (ΔSmO2-I or ΔSmO2-IM) instead of (ΔSmO2-m.intercostales).
You could remove the abbreviations used in the abstract.
Introduction
Page 2; line 71: I am aware that the abbreviation “RR and Vt” is widely used in the field of exercise sciences. However, it should be explained the first time that it appears in the text.
Page 2; line 75: Idem for “VT1 and VT2”.
The authors must highlight why this investigation is so important?
Materials and Methods
Page 2; line 82: Please, justify your sample size.
Page 2; line 84: change [BMI] for (BMI).
Page 2; line 84: “VO2-peak”. It should be explained the first time that it appears in the text.
Design of study and Participants
Was the study approved by the ethics committee? And were the ethical principles established in the Declaration of Helsinki taken into account? Please report it.
It would be interesting to report the marks of the participants in the marathon.
Page 3; line 103: change “peak oxygen consumption (VO2-peak)” for “VO2-peak”.
Page 3; line 112: “STPD”. It should be explained the first time that it appears in the text.
Measurement of SmO2
Please, could the authors report more information about the position of MOXY in the intercostal zone?
The NIRS was always placed on the right side?
As the authors say, there is little evidence on the oxygenation of the intercostal muscles. This is mainly due to the difficulty of positioning the device to obtain the data.
Do the authors have data on the reliability and reproducibility of the measurements? Please report them.
Page 4; line 159-184: The explanation of these paragraphs can be a bit confusing for the reader. The authors could rewrite these sections.
Data analysis
Page 4; line 169-170: Were SmO2 and VO2 synchronized? How? Please, could the authors add more details?
Conclussions
Page 11; line 347-355: These lines could be removed, because that conclusion was not the purpose of this study.
Round 2
Reviewer 2 Report
Authors have addressed all my concerns.
Before final acceptance, authors should specify the mean personal best result in marathon of the subjects. Author indicate that the subjects are high-level runners but do not indicate the time in Marathon.
The article can be accepted after that correction
